# Effect of Sewage Sludge Addition on Microstructure and Mechanical Properties of Kaolin-Sewage Sludge Ceramic Bricks

Xuan Zhang [1,2,3], Yang Jiao [1,2,3], Laihao Yu [4], Lili Liu [1,2,3], Xidong Wang [1,2,3],* and Yingyi Zhang [4],*

1   School of Materials Science and Engineering, Peking University, Beijing 100871, China;
    zhangxuanpku@pku.edu.cn (X.Z.); jiaoyang518@foxmail.com (Y.J.); liu-0806@163.com (L.L.)
2   Department of Energy and Resources Engineering, College of Engineering, Peking University, Beijing 100871, China
3   Beijing Key Laboratory for Solid Waste Utilization and Management, Peking University, Beijing100871, China
4   School of Metallurgical Engineering, Anhui University of Technology, Maanshan 243002, China;
    aa1120407@126.com
*   Correspondence: xidong@pku.edu.cn (X.W.); zhangyingyi@cqu.edu.cn (Y.Z.)

**Abstract:** The dramatic increase in sewage sludge production requires researchers to develop and explore more commercially viable ways for alleviating current environmental and socioeconomic challenges connected with its routine management. It has been established that sewage sludge can be processed to fabricate various valuable products or as fuels for electricity generation. In this research, kaolin (calcined from coal gangue) and sewage sludge were successfully used to prepare porous ceramic bricks without any additives. The effect of sewage sludge on the microstructure, phase composition, and mechanical properties of kaolin-sewage sludge ceramic bricks was investigated. The results show that the kaolin-sewage sludge ceramic bricks are mainly composed of mullite ($3Al_2O_3 \cdot 2SiO_2$), sillimanite ($Al_2SiO_5$), aluminum phosphate ($AlPO_4$), hematite ($Fe_2O_3$) as well as a small amount of quartz ($SiO_2$). The ceramic bricks present a typical porous structure, and the number and size of micropores increases noticeably with the increase of sewage sludge content. The sintering shrinkage rate and porosity of ceramic bricks increased significantly with the increase of sewage sludge content, which is mainly attributed to the increase of liquid phase proportion and high temperature volatilization. Sewage sludge can significantly improve the mechanical properties of kaolin-sewage sludge ceramic bricks. When the sewage sludge content is 30 wt.%, the ceramic bricks present the maximum compressive strength and flexural strength and high porosity (32.74%). The maximum sintering shrinkage rate and porosity are 12.17% and 40.51%, respectively.

**Keywords:** sewage sludge; ceramic bricks; microstructure; mechanical properties; porosity; shrinkage

## 1. Introduction

With the development of urbanization and rapid population growth, over 60 million tons of sewage sludge are produced every year in the world. Furthermore, a large amount of sewage sludge is stockpiled and landfilled, resulting in serious waste of land resources and environmental pollution [1–3]. Previous studies have pointed out that the main components of sewage sludge are $Al_2O_3$, $SiO_2$, $P_2O_5$, and $Fe_2O_3$, and the trace elements such as Pb, Cr, and Ni all exceed the limit of environmental requirements [4–7]. Comprehensive utilization of solid waste from sewage sludge is an effective way to avoid secondary pollution [8,9].

At present, sewage sludge is mainly used as agricultural fertilizer (phosphate fertilizer) or soil remediation, as fuel for power plants, and as raw materials for the preparation of building materials and ceramic materials [10–14]. It is worth noting that the direct incineration of sewage sludge will cause serious corrosion to the equipment, and especially the incineration of high-water sludge will consume a lot of energy [15–17]. In addition, sludge incineration will also cause the enrichment of heavy metals in dust and secondary pollution, and the sludge residue of combustion is directly accumulated or buried, which seriously pollutes the surrounding soil [18,19]. However, the content of $Al_2O_3$ and $SiO_2$ in

sewage sludge and its calcination residue is high, which can be used as additives for ceramic materials [20–23]. Therefore, sewage sludge can be used to produce clay bricks, refractory materials, foam ceramics, glass ceramics, etc. [24–27]. Hegazy et al. [28] considered that the mineral composition of sewage sludge and clay was similar and prepared clay bricks with good performance by using 50% sewage sludge, 25% rice husk, and 25% silica ash. Qi et al. [29] prepared a kind of porous ceramic with a sewage sludge content of 50 wt.% by using coal gangue and sewage sludge. Sewage sludge has good sintering and pore-forming function, which significantly improves the sintering characteristics of coal gangue.

Moreover, ceramics, bricks, or ceramsites prepared from sludge have great application potential in the field of water treatment due to its high porosity and good mechanical properties [30–33], which can be used as building thermal insulation materials, refractories, and water treatment materials [34–36]. For example, Tian et al. [30] used sewage sludge to prepare novel glass-ceramics through the microwave heating method, which can not only reduce energy consumption, but also significantly improve the structure and performance of ceramics. Zhou et al. [19] proposed a production method of porous thermal insulation bricks based on municipal sewage sludge to implement the reuse of solid waste and found that excessive municipal sewage sludge would result in the deterioration of brick strength. Chen et al. [17] developed porous ceramsites with good adsorption performance for Pb (II) by using co-combustion ash of various solid wastes containing sewage sludge, which could effectively treat Pb (II) polluted wastewater. Therefore, the utility value of urban sludge is high, which has important social and economic benefits for the comprehensive utilization and application of sewage sludge [37,38].

In this work, kaolin (calcined from coal gangue) and sewage sludge were successfully used to prepare kaolin-sewage sludge ceramic bricks by sintering at 1250 °C. The effect of sewage sludge content on the microstructure and phase composition were investigated. In addition, the effects of sludge content and microstructure on the sintering characteristics, porosity, and mechanical properties of ceramic bricks were also discussed. It should be noted that the content of sewage sludge in this study was controlled at 10–40 wt.%, namely, the performance and structure of sludge-free samples were not analyzed because they were almost unconsolidated or developed due to the high melting point of kaolin when sintered at 1250 °C.

## 2. Experimental Procedure

### 2.1. Materials

The raw materials used in the experiment were kaolin and sewage sludge. Sewage sludge came from the No. 1 sewage treatment plant in Yangquan, Shanxi, China. It had a high water content, and the weight loss rate after drying was 63%. Kaolin with fire resistance of 1750 °C was calcined from the coal gangue of Shanxi Huayang New Material Group (No. 2 Coal Mine). Raw materials were dried by a drying oven at 120 °C for 10 h and then were milled by a high-energy ball mill (YXQM-4L) at the speed of 1500 r·min$^{-1}$ for 2 h. The chemical compositions of sewage sludge and kaolin were determined by X-ray fluorescence (XRF) and are presented in Table 1. It can be seen that the main composition of sewage sludge was $Al_2O_3$ (15.42 wt.%), $SiO_2$ (34.84 wt.%), $Fe_2O_3$ (12.37 wt.%), and $P_2O_5$ (18.65 wt.%). It has been established that $SiO_2$ in sewage sludge mostly exists in the amorphous form because of low crystallinity, which is beneficial to the insulation performance of ceramic bricks. Furthermore, the main composition of kaolin was $Al_2O_3$ (41.53 wt.%) and $SiO_2$ (53.82 wt.%). In addition, there were small amounts of toxic and harmful substances in sewage sludge, such as $Cr_2O_3$ (0.21 wt.%) and ZnO (0.25 wt.%). It can also be seen from Table 1 that the loss on ignition (LOI) of sewage sludge and kaolin was 37.64% and 15.21%, respectively, revealing that the content of organic components was high, which is conducive to the formation of pores in ceramic bricks.

**Table 1.** Chemical compositions of sewage sludge and kaolin determined by XRF (wt.%).

| Material | $Al_2O_3$ | $SiO_2$ | $Fe_2O_3$ | $P_2O_5$ | CaO | MgO | $K_2O$ | $Na_2O$ | $TiO_2$ | $Cr_2O_3$ | ZnO | Other | LOI |
|---|---|---|---|---|---|---|---|---|---|---|---|---|---|
| Sewage sludge | 16.42 | 34.84 | 12.37 | 18.65 | 4.26 | 2.07 | 2.75 | 1.61 | 0.47 | 0.21 | 0.25 | 6.56 | 37.64 |
| Kaolin | 42.53 | 53.82 | 0.73 | - | - | - | - | - | 1.24 | - | - | 1.68 | 15.21 |

*2.2. Preparation of Ceramic Bricks*

The raw materials were weighed by an electronic balance with an accuracy of $10^{-4}$ g, the specific proportion of sludge and kaolin was evenly mixed, and the methylcellulose solution with a concentration of 5 wt.% was used as the binder. It is worth mentioning that the methylcellulose is easily decomposed to produce gas at high temperature, which also contributes to the formation of pores in sintered ceramic bricks. In this work, the additional proportions of sludge were 10, 20, 30, and 40 wt.%, respectively. Furthermore, the proportion of the mixture is shown in Table 2. The prepared mixture was pressed into a rectangular green brick of $50 \times 15 \times 8$ mm$^3$ by an isostatic pressing molding machine under the pressure of 20 MPa and then dried by a drying oven at 120 °C for 10 h. Finally, the ceramic bricks were sintered through a high-temperature muffle furnace at 1250 °C for 40 min and then cooled to room temperature with the furnace. Note that the high sintering temperature was due to the very high melting point of kaolin, which was also conducive to saving sintering time.

**Table 2.** Designed proportions of kaolin-sewage sludge ceramic bricks (wt.%).

| Sample | S1 | S2 | S3 | S4 |
|---|---|---|---|---|
| Sewage sludge | 10 | 20 | 30 | 40 |
| Kaolin | 90 | 80 | 70 | 60 |
| Binder (methylcellulose solution) | | 10 | | |

*2.3. Characterization Techniques*

The phase composition was identified by X-ray diffractometry (XRD, D8-Advance, Bruker, Germany) with Cu radiation (λ = 1.5406 Å) from 10 to 90°. The microstructure and elemental distribution of the kaolin-sewage sludge ceramic bricks were examined by scanning electron microscopy and energy-dispersive spectroscopy (SEM–EDS, JSM-6510LV, JEOL, Tokyo, Japan). The compressive strength and flexural strength were measured by a universal mechanical testing machine. Forty samples were used to measure the average values of sintering shrinkage, compressive strength, and flexural strength of ceramic bricks. The implementation standard of compressive strength and flexural strength were GB/T 8489-2006 and GB/T 6569-2006, respectively. The sintering shrinkage rate was calculated by the following formula [23,32]:

$$L_v = \frac{1}{3} \sum_{i=1}^{3} \frac{(L_i - L'_i)}{L_i} \times 100\%$$

where $L_v$ is the sintering shrinkage rate (%), $L_i$ is the length before sintering (mm), and $L_i'$ is the sample length after sintering (mm).

**3. Results and Discussion**

*3.1. Phase Composition of Ceramic Bricks*

The effect of sewage sludge content on the phase evolution of the kaolin-sewage sludge ceramic bricks is shown in Figure 1. It can be seen that the kaolin-sewage sludge ceramic bricks were mainly composed of mullite ($3Al_2O_3 \cdot 2SiO_2$), sillimanite ($Al_2SiO_5$), aluminum phosphate ($AlPO_4$), and hematite ($Fe_2O_3$), as well as a small amount of quartz ($SiO_2$). With the increase of sewage sludge content, the diffraction peak intensity of mullite and quartz phases decreased slightly, which was mainly due to the decrease of $Al_2O_3$

and $SiO_2$ content in ceramic bricks. In addition, a small amount of the $Fe_2O_3$ phase was also observed, which mainly came from sludge. The reduction of the mullite phase was helpful to improve the sintering performance of ceramic bricks, and the high temperature decomposition of sewage sludge also promoted the formation of porous ceramic tiles.

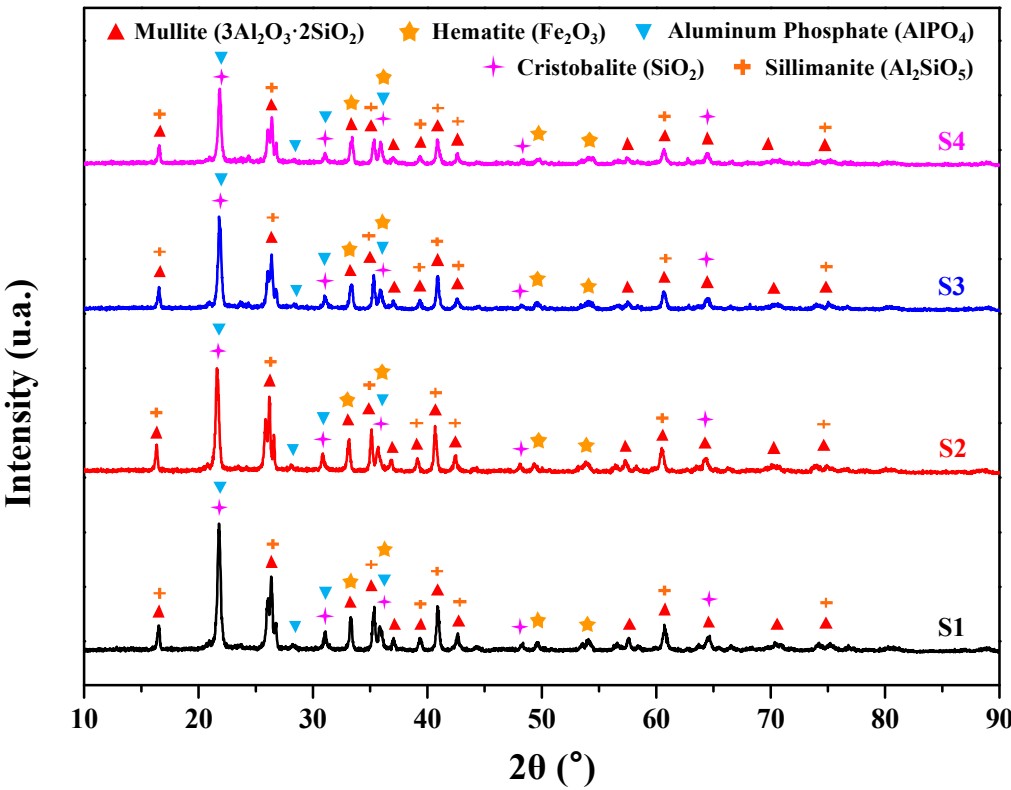

**Figure 1.** XRD patterns of the ceramic bricks with various sewage sludge contents sintered at 1250 °C for 40 min.

### 3.2. Microstructure of Ceramic Bricks

Figure 2 shows the surface morphology of ceramic bricks with various sewage sludge contents sintered at 1250 °C for 40 min. The elemental composition of micro-zones is shown in Table 3. The SEM–EDS results indicated that the kaolin-sewage sludge ceramic bricks were mainly composed of light gray mullite ($3Al_2O_3 \cdot 2SiO_2$), dark gray sillimanite ($Al_2SiO_5$), white $Fe_2O_3$ phase, and black $SiO_2$ phase, and the analysis of EDS was consistent with the results of XRD. However, no heavy metal elements were observed in ceramic bricks, indicating that heavy metal elements in sewage sludge were effectively diluted and solidified. It can be seen from Figure 2a–c that when the content of sewage sludge was 10 wt.%, the sintering performance of the kaolin-sludge ceramic bricks was extremely poor. The surface structure of ceramic bricks was very loose, which presented an obvious granular structure because the sewage sludge content in the S1 sample was too low to produce a glassy liquid phase, resulting in unconsolidated or a relatively low degree of consolidation for ceramic bricks. When the content of sewage sludge increased to 20–40 wt.%, the glassy liquid phase began to form to significantly improve the sintering properties of ceramic bricks. As shown in Figure 2d,g,j, with the increase of sludge content, the number and size of micropores increased noticeably, and the increase of sintering liquid phase greatly promoted the bond between particles, as shown in Figure 2e,h,k. In addition, some disseminated white $Fe_2O_3$ particles with sizes of 1–2 μm were also observed when the contents of sewage sludge were 20–40 wt.%, as shown in Figure 2c,f,i,l. The

reaction sintering and pore forming mechanism of sewage sludge can be mainly described by Equations (1)–(4).

$$Organics + O_2(g) \rightarrow CO_2(g) + H_2O(g) \tag{1}$$

$$Al_2O_3 + 2SiO_2 = 3Al_2O_3 \cdot 2SiO_2 \tag{2}$$

$$Al_2O_3 + SiO_2 = Al_2O_3 \cdot SiO_2 \tag{3}$$

$$Al_2O_3 + P_2O_5 = 2AlPO_4 \tag{4}$$

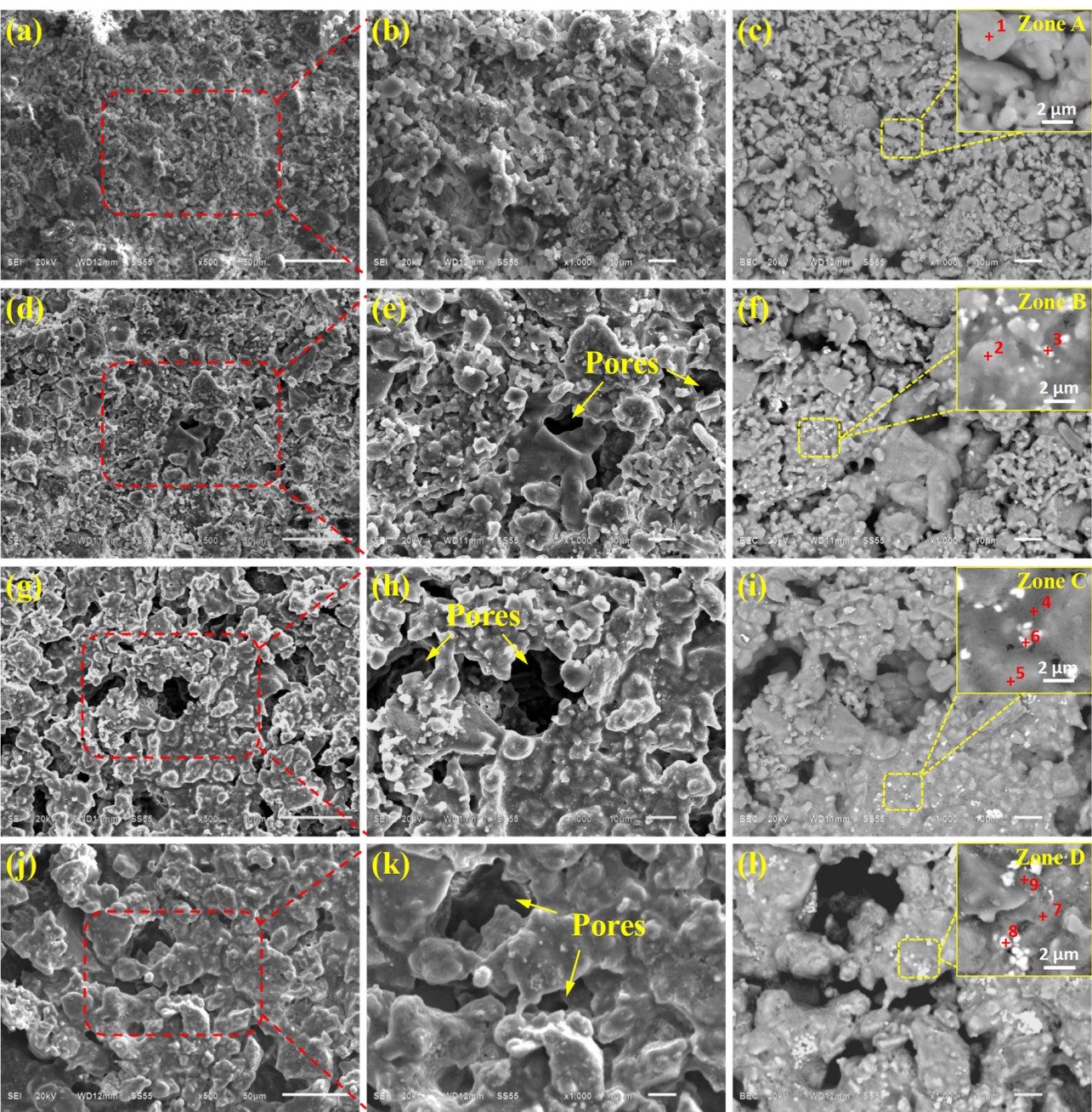

**Figure 2.** Surface morphology of ceramic bricks with various sewage sludge contents sintered at 1250 °C for 40 min. (**a**–**c**) S1—10 wt.%; (**d**–**f**) S2—20 wt.%; (**g**–**i**) S3—30 wt.%; (**j**–**l**) S4—40 wt.%.

**Table 3.** Elemental chemical composition of various points in micro-zones A–D from Figure 2.

| No. | Element Composition (wt.%) | | | | | | | | | | | Mineral Phases |
|---|---|---|---|---|---|---|---|---|---|---|---|---|
| | O | Al | Si | Ca | Mg | Ti | Fe | Zr | P | Na | K | |
| 1 | 68.85 | 19.73 | 16.06 | - | - | - | 0.56 | 0.79 | - | - | - | Mullite ($3Al_2O_3 \cdot 2SiO_2$) + Quartz ($SiO_2$) |
| 2 | 63.34 | 14.38 | 14.16 | - | - | - | 1.38 | - | 1.05 | - | - | Mullite ($3Al_2O_3 \cdot 2SiO_2$) + Quartz ($SiO_2$) |
| 3 | 45.26 | 9.38 | 8.29 | 0.64 | - | 0.47 | 34.30 | 1.65 | - | - | - | Hematite ($Fe_2O_3$) |
| 4 | 63.34 | 11.24 | 18.12 | 1.40 | 0.45 | - | 1.76 | - | 1.84 | 0.91 | 0.93 | Sillimanite ($Al_2SiO_5$) + Quartz ($SiO_2$) |
| 5 | 69.56 | 13.37 | 12.28 | 0.57 | 0.44 | - | 1.19 | 0.74 | 0.60 | 0.88 | 0.37 | Mullite ($3Al_2O_3 \cdot 2SiO_2$) + Quartz ($SiO_2$) |
| 6 | 46.31 | 9.43 | 8.21 | 0.72 | - | 0.45 | 34.35 | 1.63 | - | - | - | Hematite ($Fe_2O_3$) |
| 7 | 46.51 | 21.09 | 24.38 | 0.97 | - | - | 5.14 | 1.90 | - | - | - | Sillimanite ($Al_2SiO_5$) + Quartz ($SiO_2$) |
| 8 | 42.83 | 8.81 | 8.19 | - | - | 1.54 | 36.42 | 2.20 | - | - | - | Hematite ($Fe_2O_3$) |
| 9 | 53.37 | 21.45 | 18.27 | 1.30 | 0.43 | - | 1.73 | - | 1.81 | 0.80 | 0.63 | Mullite ($3Al_2O_3 \cdot 2SiO_2$) + Quartz ($SiO_2$) |

Chen et al. [39] reported that the organic materials in sewage sludge are mainly composed of carbohydrates, proteins, and lignin. It can be seen that the combustion of organic matter not only provided additional heat for the sintering reaction but also was the main reason for the formation of the porous structure. In addition, the high content of $SiO_2$ and $P_2O_5$ in sewage sludge promoted the formation of silimanite ($Al_2SiO_5$) and aluminum phosphate ($AlPO_4$) phases. As is well-known, the physical and chemical properties of sillimanite ceramics are noticeably better than mullite ceramics. Moreover, aluminum phosphate is also an efficient fluxing agent, which can significantly improve the sintering properties of ceramic materials [40,41].

Figure 3 shows the cross sectional LSCM images of ceramic bricks with various sewage sludge contents. Laser color images show that the color of ceramic tiles gradually changed from white to yellow with the increase of sludge content, and the area with high sludge content was obviously black, as shown in Figure 3a–d. The LSCM intensity images showed that with the increase of the sewage sludge content, the number and size of cross section micropores increased significantly, as shown in Figure 3e–h. LSCM 3D images showed that the cross-sectional morphology was very flat when the sewage sludge content was 10 wt.%, and the values of Ra, Rq, and Rz were only 3.269, 3.956, and 43.341 µm, respectively. This was mainly due to the brittle fracture of ceramic brick. As shown in Figure 3j–l, the height fluctuation of cross-sectional morphologies increased noticeably, and the roughness values also increased noticeably. The maximum roughness value was observed when the sewage sludge content was 40 wt.%, and the values of Ra, Rq, and Rz were 9.305, 11.525, and 123.284 µm, respectively. This was mainly due to the increase of flexural strength and porosity.

### 3.3. Mechanical Properties of Ceramic Bricks

Figures 4–6 present the apparent density, sintering shrinkage rate, and porosity of the ceramic bricks with various sewage sludge contents. It can be seen that the density of ceramic bricks before and after sintering had the same change trend with the increase of sewage sludge content; that is, it first increased by about 15%–20% and then decreased slightly, which is related to the microstructure of ceramic bricks and the volatilization of sewage sludge. Moreover, with the increase of sludge content, the sintering shrinkage and porosity of ceramic bricks increased significantly, as shown in Figures 5 and 6. The maximum sintering shrinkage rate and porosity were observed when the sewage sludge content was 40 wt.%, and the maximum sintering shrinkage rate and porosity were 12.17% and 40.51%, respectively. The high temperature volatilization of sewage sludge and the increase of liquid phase proportion were the main reasons for the increase of sintering shrinkage rate and porosity. Furthermore, the extremely high porosity would destroy the microstructure of the ceramic bricks, resulting in a decrease in the strength of the S4 sample.

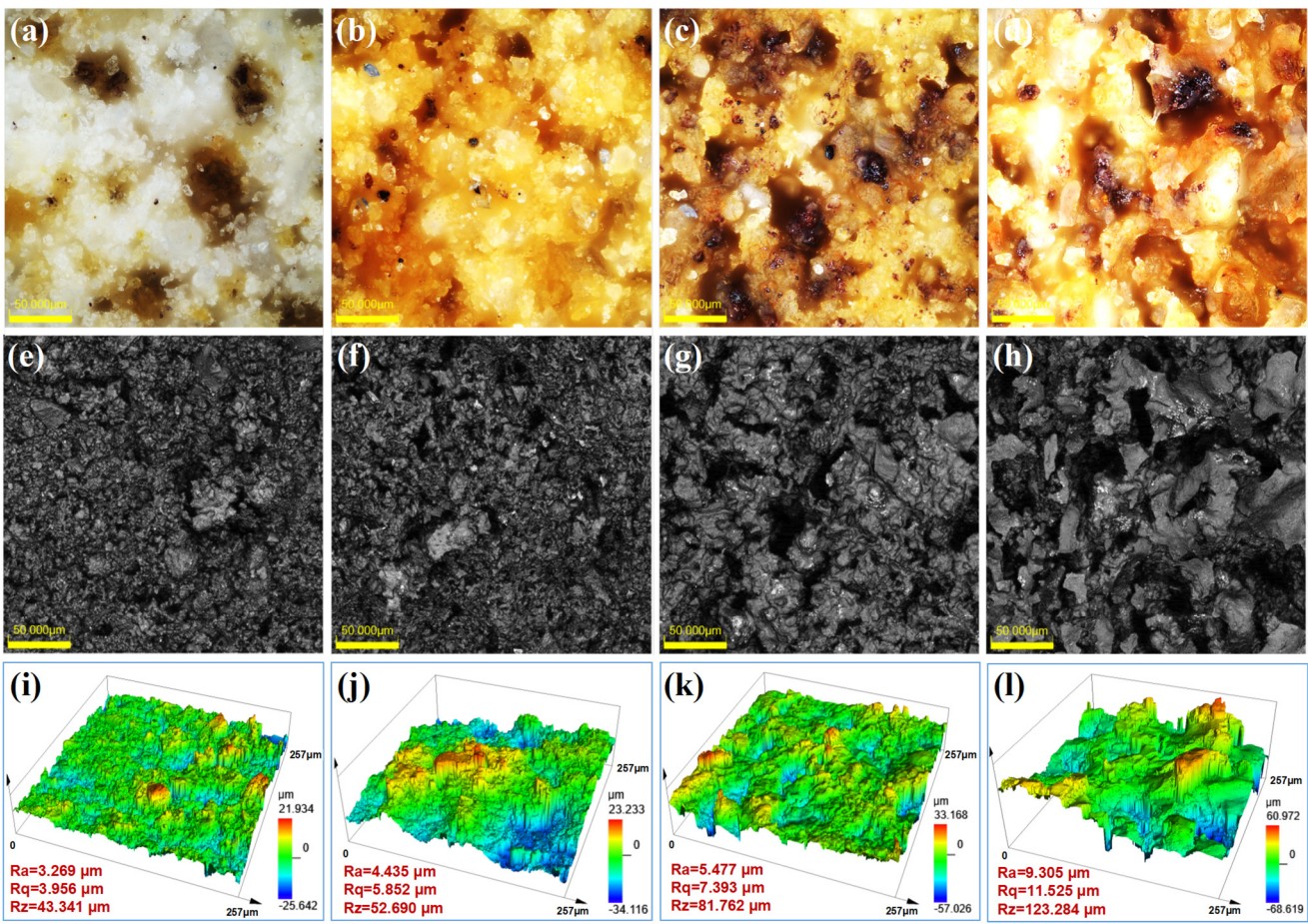

**Figure 3.** Cross sectional LSCM images of ceramic bricks with various sewage sludge contents sintered at 1250 °C for 40 min. (**a–d**) Laser color images; (**e–h**) white light optical images; (**i–l**) LSCM 3D images; (**a,e,i**) S1—10 wt.%; (**b,f,j**) S2—20 wt.%; (**c,g,k**) S3—30 wt.%; (**d,h,l**) S4—40 wt.%.

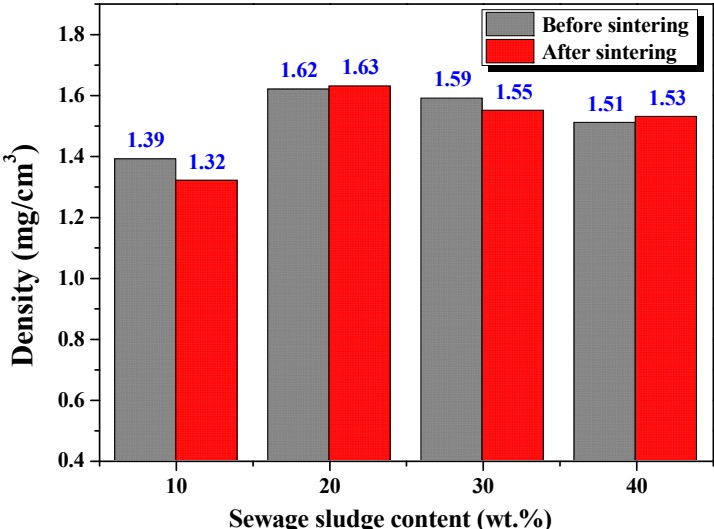

**Figure 4.** Apparent density of the ceramic bricks with various sewage sludge contents.

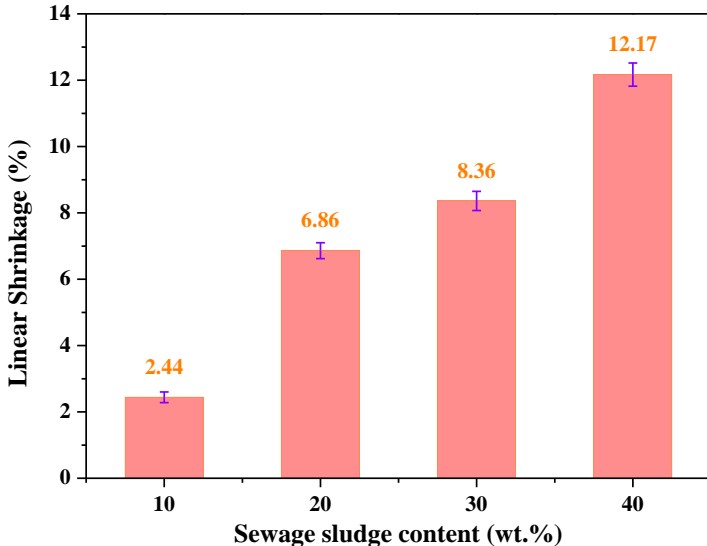

**Figure 5.** Sintering shrinkage rate of the ceramic bricks with various sewage sludge contents.

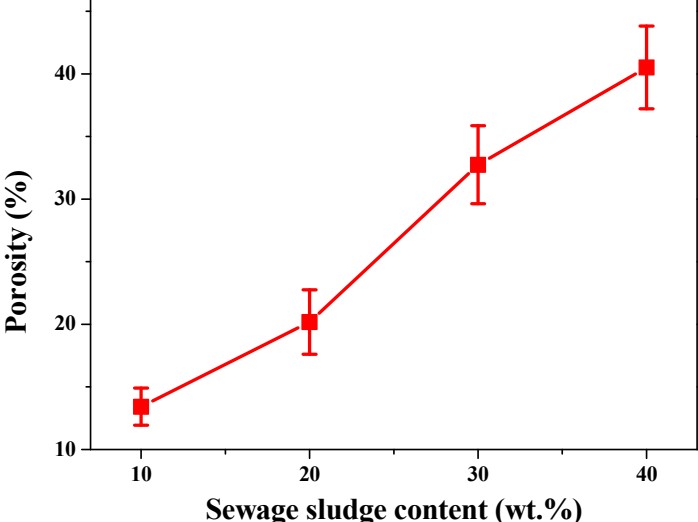

**Figure 6.** Porosity of the ceramic bricks with various sewage sludge contents.

Figure 7 shows the compressive strength and flexural strength of the ceramic bricks with various sewage sludge contents. It can be seen that the compressive strength of the kaolin-sewage sludge ceramic bricks presented a kind of gradually increasing trend with the increase of sewage sludge content. However, the flexural strength showed a law of an increase first and then a decrease. When the sewage sludge content was 10 wt.%, the compressive strength and flexural strength of ceramic bricks were only 5.12 and 3.21 MPa, respectively. As shown in Figure 2b,c and Figure 3a,e, when the sludge content was 10 wt.%, the surface and cross-section microstructure of ceramic bricks were very loose and mainly composed of granular structures, which led to the low compressive strength and flexural strength of ceramic bricks. When the sewage sludge content increased to 20 wt.%, the compressive strength and flexural strength of ceramic bricks increased significantly. The compressive strength of ceramic bricks was similar when the sewage sludge content was 30 and 40 wt.%. The maximum compressive strength and flexural strength was observed when the sewage sludge content was 30 wt.%. The formation of silimanite ($Al_2SiO_5$) and aluminum phosphate ($AlPO_4$) phases was the main reason for the improvement of sintering and mechanical properties of porous ceramics with high sludge contents. However, the flexural strength of ceramic tile decreased significantly when the sludge content was

40 wt.%, and the significant increase of the number and size of micropores was the main reason for the sharp decrease of flexural strength. Therefore, increasing the sludge content can significantly improve the sintering properties and porosity of kaolin ceramics and effectively improve its mechanical properties.

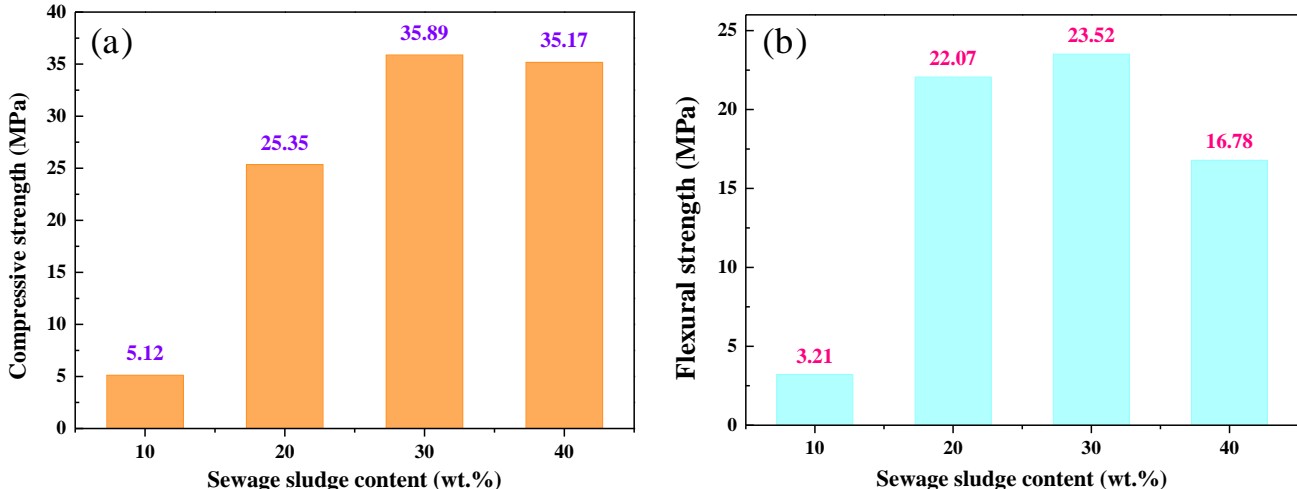

**Figure 7.** Compressive strength and flexural strength of the ceramic bricks with various sewage sludge contents. (**a**) Compressive strength; (**b**) flexural strength.

Table 4 shows the comparison of this work with other studies. The listed bricks include clay bricks, thermal insulation bricks, and ceramic bricks. The developed ceramic bricks presented excellent compressive strength characteristics compared with other sludge-based sintered bricks. Moreover, the porosity and density of the sintered ceramic bricks in this work exhibited tremendous competitiveness in comparison with those from previous studies. Although the shrinkage performance of ceramic bricks needs to be further suppressed, it has a great potential for application in the construction field. In addition, all the raw materials of sintered ceramic bricks are derived from waste, so the production cost is low.

**Table 4.** Product properties of various sintered sludge-based bricks.

| Waste Raw Materials | Product | Density (mg/cm$^3$) | Shrinkage (%) | Porosity (%) | Compressive Strength (MPa) | Ref. |
|---|---|---|---|---|---|---|
| Sewage sludge; pure clay body | Clay bricks | - | 0.88–1.07 | - | 6.20–17.85 | [3] |
| Sewage sludge; oven slag; fly ash | Clay bricks | 2.14–3.38 | 2.82–10.98 | 11.87–44.68 | 2.26–31.67 | [13] |
| Municipal sewage sludge; clay; rice husk ash; Na$_2$CO$_3$ | Thermal insulation bricks | 1.25–1.55 | - | 36.19–44.31 | 4.71–13.59 | [18] |
| Municipal sewage sludge; clay; silica fume; fly ash | Thermal insulation bricks | 1.34–1.55 | - | 34.20–41.45 | 5.90–23.50 | [19] |
| Sewage sludge; kaolin | Ceramic bricks | 1.32–1.63 | 2.44–12.17 | 13.27–40.51 | 5.12–35.89 | This work |

## 4. Conclusions

In this work, kaolin and sewage sludge were successfully used to prepare porous ceramic bricks without any additives. The effect of sewage sludge on the microstructure, phase composition, and mechanical properties of kaolin-sewage sludge ceramic bricks were investigated. The detailed experimental conclusions are as follows:

(1) The kaolin-sewage sludge ceramic bricks are mainly composed of mullite ($3Al_2O_3 \cdot 2SiO_2$), sillimanite ($Al_2SiO_5$), aluminum phosphate ($AlPO_4$), and hematite ($Fe_2O_3$), as well as a small amount of quartz ($SiO_2$). The ceramic bricks present a typical porous structure, and the number and size of micropores increase noticeably with the increase of sewage sludge content.

(2)   The sintering shrinkage rate and porosity of ceramic bricks increase significantly with the increase of sewage sludge content, which is mainly attributed to the increase of the liquid phase proportion and high temperature volatilization.

(3)   Sewage sludge can significantly improve the mechanical properties of kaolin-sewage sludge ceramic bricks. When the sewage sludge content is 30 wt.%, the ceramic bricks present the maximum compressive strength and flexural strength with a porosity of 32.74%. The maximum sintering shrinkage rate and porosity are 12.17% and 40.51%, respectively.

(4)   The formation of silimanite ($Al_2SiO_5$) and aluminum phosphate ($AlPO_4$) phases is the main reason for the improvement of the sintering and mechanical properties of porous ceramics with high sludge content. Increasing the sludge content can significantly improve the porosity and mechanical properties of ceramic bricks.

**Author Contributions:** The manuscript was written through contributions of all authors. Performed the Resources, Writing—Review and Editing, Supervision, and Data Curation, X.W. and Y.Z.; Performed the SEM, EDS, and LSCM measurements, X.Z. and Y.J.; Prepared and performed samples, and image processing, L.Y. and L.L. All authors have read and agreed to the published version of the manuscript.

**Funding:** This work was supported by the National Key Research and Development Program of China (2017YFB0603800 and 2017YFB0603802).

**Institutional Review Board Statement:** Not applicable.

**Informed Consent Statement:** Not applicable.

**Data Availability Statement:** Not applicable.

**Conflicts of Interest:** The authors declare no conflict of interest.

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
