# Peer review of "Effect of Sewage Sludge Addition on Microstructure and Mechanical Properties of Kaolin-Sewage Sludge Ceramic Bricks"

_coatings, doi:10.3390/coatings12070944_

Round 1

Reviewer 1 Report

Dear Authors! The article is dedicated to the urgent problem of wate recycling with the production of ceramic bricks. However, there are some arguable points:
1. Authors didn't describe any amorphous glassy phases, SiO2 or P2O5-based. The absence of glassy phase is doubtful, it should be checked.
2. The aim of the article (final paragtaph before 2. Experimental procedure) is declared as "to prepare !porous! ceramic bricks". It doesn't correspond to the title, so the title or the aim should be corrected.
3. LOI column should be added to the Table 1, as long as it is the main source of pore formation.
4. Also, effect of the methylcellulose binder decomposition on the pore formation should be described.
5. Why did authors choose 1250 C as the sintering temperature? Especially when this temperature is insufficient for the sintering of low-sludge samples?
After answering these questions, the article can be published in the Journal.

Reviewer 2 Report

Title: OK

Abstract: From the abstract, it seems that there is no comparison with control ceramic bricks containing no sludge.

Keywords: Porosity and shrinkage may be added. Bricks and recycling are absent from the keywords.

Introduction: The literature review is brief and needs similar studies along with the research gaps. Also, there is a need to highlight novelty.

Experimental procedure:

1.       Line 69: Is it not expensive to dry the materials at 120ºC. Was it done for gaining economy in time (10 hours)?  

2.       Table 1: Kaolin does not have any lime content? Please confirm. The chemical composition of sludge is more useful for ceramic operations than the main constituent Kaolin. This is strange.

3.       Line 85-86: The sintering temperature is high and the duration is only 40 minutes. Kindly explain the choice.

Results and discussion:

1.       Line 98-99: Old standards are used. Why?

2.       Figure 2: Kindly change “2” with “S2”.

3.       Line 143: Kindly avoid “we”, “you”, and “I” throughout the manuscript.

4.       Table 2: What do the serial numbers “1-9” present?

5.       Line 177: Porosity is extremely high (40%) for S4. Are there any consequences?

6.       Line 171-173: Kindly correct your sentence. The density first increases from 10-20% and then decreases. Correct your sentence accordingly with reasoning.

7.       Figure 4, Figure 5 and Figure 6 do not support each other. Although Figure 5 and Figure 6 do support each other. If there is an increase in porosity with sludge content, there should be a decrease in density. This does not happen in your case.

8.       Figure 7 needs attention. The flexural to compressive strength ratio for S10 is 88%, which is very surprising.

9.       Increase of porosity with an increase in compressive strength is another result from this study, which needs some deliberation.  A high increase in porosity, as well as shrinkage, is accompanied by very high strength also!!!

Conclusions: OK

Overall:

1.       There is one thing, which needs deliberation. The main material i.e. Kaolin is deficient in hematite and lime, still, it is considered suitable for making ceramic bricks. Whereas the supplementary material i.e. sludge contains almost good proportions of all the oxides that are supportive to brick manufacturing. The supplementary material is more useful than the main ingredient!!!

2.       In research, the results are interlinked to each other as well as to the previous studies. In the current work, there is a scarcity of inter-linkage.

3.       A study containing 100% kaolin is absent. 

Reviewer 3 Report

Manuscript title: Effect of sewage sludge addition on microstructure and mechanical properties of kaolin-sewage sludge ceramic bricks

1-      Problem statement/Introduction is missing in the abstract.

2-      More literature should be added into the introduction.

3-      Show the research gap the end of the introduction.

4-      More discussion is required for section 3.3.

5-      Compare the results with the conventional and existing bricks in the literature.

Round 2

Reviewer 2 Report

Accept in present form

Reviewer 3 Report

Addressed accordingly.